# Sex Differences in the Anti-Hypertensive Effect of Calcium-Channel Blockers: A Systematic Review and Meta-Analysis

**DOI:** 10.3390/biomedicines11061622

**Published:** 2023-06-02

**Authors:** Eveline M. van Luik, Esmée W. P. Vaes, Maud A. M. Vesseur, Nick Wilmes, Daniek A. M. Meijs, Sophie A. J. S. Laven, Zenab Mohseni-Alsalhi, Sander de Haas, Marc E. A. Spaanderman, Chahinda Ghossein-Doha

**Affiliations:** 1Department of Obstetrics and Gynaecology, Maastricht University Medical Center (MUMC+), 6229 HX Maastricht, The Netherlands; 2Cardiovascular Research Institute Maastricht, School for Cardiovascular Diseases, Maastricht University, 6229 ER Maastricht, The Netherlands; 3Department of Obstetrics and Gynaecology, Radboud University Medical Center, 6525 GA Nijmegen, The Netherlands; 4Department of Cardiology, Maastricht University Medical Center (MUMC+), 6229 HX Maastricht, The Netherlands

**Keywords:** hypertension, cardiovascular disease, calcium-channel blockers, sex differences, systematic review, meta-analysis

## Abstract

Cardiovascular disease (CVD) is the number one cause of death worldwide, with hypertension as the leading risk factor for both sexes. As sex may affect responsiveness to antihypertensive compounds, guidelines for CVD prevention might necessitate divergence between females and males. To this end, we studied the effectiveness of calcium channel blockers (CCB) on blood pressure (BP), heart rate (HR) and cardiac function between sexes. We performed a systematic review and meta-analysis on studies on CCB from inception to May 2020. Studies had to present both baseline and follow-up measurements of the outcome variables of interest and present data in a sex-stratified manner. Mean differences were calculated using a random-effects model. In total, 38 studies with 8202 participants were used for this review. In females as compared to males, systolic BP decreased by −27.6 mmHg (95%CI −36.4; −18.8) (−17.1% (95%CI −22.5;−11.6)) versus −14.4 mmHg (95%CI −19.0; −9.9) (−9.8% (95%CI −12.9;−6.7)) (between-sex difference *p* < 0.01), diastolic BP decreased by −14.1 (95%CI −18.8; −9.3) (−15.2%(95%CI −20.3;−10.1)) versus −10.6 mmHg (95%CI −14.0; −7.3) (−11.2% (95%CI −14.8;−7.7)) (between-sex difference *p* = 0.24). HR decreased by −1.8 bpm (95%CI −2.5; −1.2) (−2.5% (95%CI −3.4; −1.6)) in females compared to no change in males (0.3 bpm (95% CI −1.2; 1.8)) (between-sex difference *p* = 0.01). In conclusion, CCB lowers BP in both sexes, but the observed effect is larger in females as compared to males.

## 1. Introduction

Raised blood pressure primarily antedates cardiovascular disease (CVD), the number one cause of death worldwide [1,2]. As such, effective treatment of hypertension represents a key strategy for reducing cardiovascular diseases [2]. Pharmacological treatment and modulation of behavioral risk factors, amongst smoking cessation, weight reduction, physical activity, reduction in alcohol use and a healthy diet, lower the development of hypertension and CVD [1].

While in the past more males died from CVD, nowadays females have surpassed the other sex [3]. Females are considered more protected against cardiovascular events during the fertile period, but this protection diminishes after menopause. Consequently, CVD is a major cause of death in females above 65 years of age. However, in the younger population, gender-related factors, such as psychological stress and low socioeconomic status, are expected to have a significant influence on vascular ageing [4]. Timely treatment reduces the risk of CVD and its mortality for both sexes [5]. Weighing sex differences in risk factors such as smoking, system-biology, clinical manifestations, treatment effects and outcomes of CVD in guidelines may contribute to improved outcomes in prevention of CVD [6,7,8,9,10,11].

The predominantly used antihypertensive compounds are calcium-channel blockers (CCB), angiotensin-converting enzyme inhibitors (ACEI), angiotensin-receptor blockers (ARB), beta blockers (BB) and diuretics (DIU). CCB are one of the most widely used classes of antihypertensive agents. All approved CCB types exert their effects as pharmacologic agents inhibiting transmembrane calcium inflow through calcium channels, reducing actin–myosin interaction and the subsequent contraction of myocytes and with it, its vascular smooth muscle tone, thereby attributing to reduction in blood pressure. A remarkable number of clinical trials has investigated and proven the sufficient haemodynamic effect of CCB on cardiovascular and haemodynamic variables [12,13,14]. However, almost none of these trials have investigated the effects in females and males separately [15]. This raises the question whether CCB are equally effective in both sexes and whether treatment strategies should differentiate between both sexes. 

To this end, we studied in a systematic review and meta-analysis the acute (0–14 days), sub-acute (15–30 days) and chronic (>30 days) intervention effects of CCB’s treatment on cardiovascular and haemodynamic variables in female versus male adults diagnosed with hypertension. The variables of primary interest are systolic blood pressure (SBP) and diastolic blood pressure (DBP), mean arterial blood pressure (MAP) and heart rate (HR). Variables of secondary interest are cardiac output (CO), left ventricular ejection fraction (LVEF) and left ventricular mass (LVM).

## 2. Materials and Methods

### 2.1. Literature Search

An extensive systematic literature search was conducted on articles evaluating the effects of antihypertensive medication on cardiovascular and haemodynamic variables using PubMed (NCBI) and Embase (Ovid) databases. PubMed and Embase provided publications published from inception to May 2020. The search terms are presented in Appendix A. The search strategy aimed at studying the effect of the five antihypertensive drugs (CCB, ACEI, ARB, BB and DIU) on blood pressure, cardiac function and geometry. For this review, the articles reporting on CCB were used. The search limits used were ‘humans’ and ‘journal article’. The search served to study the following objectives:To study differences and similarities between males and females in the effect of antihypertensive medication on cardiac function and structure.To determine the representation of females in studies on the effect of antihypertensive drugs on CVD for the past century.

### 2.2. Guidelines

PRISMA 2020 reporting guidelines were taken into account. Our review was registered in the PROSPERO database with registration number CRD42021273583.

### 2.3. Eligibility Criteria

Studies had to focus on acute, sub-acute and/or chronic therapy with at least one type of CCB in female and/or male adults (≥18 years) diagnosed with hypertension. Moreover, studies had to include the mean with standard deviation (SD), standard error (SE), or 95% confidence interval (95% CI) of the baseline and follow-up measurements of one of the predefined variables (SBP, DBP, MAP, HR, CO, LVEF, and/or LVM). Studies also had to report the mean dose or dose range and treatment duration. Finally, the antihypertensive treatment had to be compared to a reference group (control, placebo or antihypertensive medication group other than CCB under study). Mean values with SD were requested from the authors by email if articles presented their data differently (for example, median with interquartile range). 

### 2.4. Study Selection

After the initial search, studies were screened based on title and abstract. During this selection, other systematic reviews and meta-analyses, literature reviews, case reports, animal studies, and in vitro studies were excluded. Studies with subjects younger than 18 years and articles in another language than English or Dutch were excluded as well. The remaining studies were screened for suitability based on full-text using the eligibility criteria. Studies were excluded if they did not separate outcomes by antihypertensive medication (if participants received more than one antihypertensive medication as intervention) or did not report the treatment duration or a mean dose or dose range for the antihypertensive medication. Studies with individuals undergoing invasive operations, participants who were exercising during measurements, undergoing dialysis or chemotherapy were excluded as well. In case the articles did not stratify the data for sex, but all other eligibility criteria were met, authors from articles published after 1980 were contacted to request sex-specific data. If no contact details were found or if authors did not respond within three weeks after sending a reminder, the article was excluded. The reason for exclusion was registered for the full-text selection. Both selection steps were performed in pairs in a blinded standardized manner (title-abstract pairs: MA-EV, CD-SL, EL-DM, ZM-JW, MV-NW; full-text pairs: CD-NW, EL-MV, DM-SL, EV-JW). Discrepancies were resolved by mutual agreement.

### 2.5. Data Extraction

Study characteristics (sample size, control group, study design), anthropometric data (age, ethnicity), intervention characteristics (dose, duration, method of measurement) and effect measures (mean and SD at baseline and after CCB intervention of the predefined variables) were collected in a predesigned format. The study results were separately extracted for females and males. In this systematic review, only blood pressure data measured using non-invasive methods were extracted. For the other variables, multiple methods were allowed. Data extraction was performed by two investigators (RA, LK). This step of the process was not performed in duplicate. 

### 2.6. Quality Assessment

The included studies were assessed for quality and risk of bias using the Cochrane recommended Risk of Bias 2 (RoB2) tool [16]. Studies were scored with “Low risk of bias”, “Some concerns” or “High risk of bias” on five domains including randomization process, deviations from intended interventions, missing data, outcome measurement and data reporting. To receive an overall risk-of-bias judgement of “Low risk of bias”, all domains had to receive this judgement. To receive an overall judgment of “High risk of bias”, at least one of the domains was scored as such. All other domain score combinations would rate a study with an overall judgement of “Some concerns”. The quality assessment was performed by two reviewers (RA, LK) and differences were solved by a third independent reviewer (DM, SL).

### 2.7. Statistical Analysis

If an SE or 95% CI was reported in the article, the SD was calculated according to the Cochrane Handbook for Systematic Review of Interventions [17]. Changes in the cardiovascular and haemodynamic variables from baseline were separately analyzed for females and males using a random-effects model as described by DerSimonian and Laird [18]. Because the included studies had some variation in study population and design, the random-effects model was chosen to account for this interstudy variation [18]. Egger’s regression test for funnel plot asymmetry was conducted to test for publication bias for each cardiovascular variable [19]. The primary outcome was the mean difference and 95% CI between baseline and follow-up of the intervention, visualized in forest plots. The relative change from baseline in percentage including 95% CI was also calculated and reported in parentheses behind the mean difference in the text. The I^2^ statistic, the ratio between heterogeneity and variability, was calculated as a measure of consistency and expressed as percentage in the forest plots. I^2^ is able to distinguish heterogeneity in data from solely sampling variance [17]. Interpretation of I^2^ was based on the guidelines in the Cochrane Handbook for Systematic Review of Interventions [17]. Sources of clinical heterogeneity (CCB type, treatment duration, and dosage) and methodological heterogeneity (quality of study) were investigated by meta-regression analyses using a mixed-effects model [17]. For the meta-analyses and meta-regression analyses, the meta package in the statistical program R version 4.0.3. was used [20,21].

## 3. Results

### 3.1. Study Selection

The literature search in PubMed and Embase provided a total of 73,867 unique records after removing duplicates (Figure 1). During the first screening, 58,737 articles were excluded resulting in 15,130 articles that were assessed based on the full text. Of those articles, 14,916 met at least one exclusion criterion and were excluded. For 766 articles (5%), it was not possible to find or access the full text at the university library or online. In total, 1141 articles (8%) had an unsuitable study design. This criterion was met when, for example, only measurements were taken during exercise, or SBP and DBP were measured using an arterial catheter. In total, 1058 articles (7%) did not report original research data; these articles were reviews, for example. In 1886 articles (13%), no antihypertensives were given to the patients participating. In 2141 articles (14%), antihypertensives were given, but treatment results were not stratified by those. In total, 1949 articles (13%) were excluded because treatment results were not stratified by sex. In total, 153 articles (1%) did not have reference measurements. In total, 3864 articles (26%) did not contain any measurements of interest. In 536 articles (4%), data were not suitably reported. In 984 articles (6%), no information was provided regarding either dose, duration, or both. Finally, there were 438 articles (3%) excluded because of other complications. At the end of the selection procedure, a total of 214 articles were classified suitable for inclusion (Figure 1). Eventually, in 38 of those articles, CCB were the provided treatment.

### 3.2. Study Characteristics 

Study characteristics and anthropometric data are visualized in Table 1. Data of 8202 subjects using CCB were included in this meta-analysis, of whom 3264 (39.8%) were female. The mean age of the subjects from the included studies was 66.2 ± 8.9 (SD) years.

Thirteen studies analyzed the effects of nifedipine [22,23,24,25,26,27,28,29,30,31,32,33,34], seven of diltiazem [29,35,36,37,38,39,40], five of amlodipine [41,42,43,44,45] and nicardipine [46,47,48,49,50], three of verapamil [51,52,53] and two of nitrendipine [54,55] and felodipine [56,57]. Lercanidipine [43], mibefradil [42], isradipine [58], lacidipine [26] and gallopamil [59] all had one study reporting on them. One study reported on both amlodipine and lercanidipine [43]. Another studied amlodipine together with mibefradil [42]. Additionally, diltiazem and nifedipine were studied together [29].

SBP was studied in 17 studies [26,28,29,33,35,36,37,39,40,43,44,45,51,52,54,55,57], DBP in 15 studies [26,29,33,35,37,39,40,43,44,45,51,52,54,55,57], MAP in 4 studies [34,41,45,57], HR in 28 studies [22,23,24,25,26,27,28,29,31,32,34,35,36,39,40,41,42,45,46,47,49,50,51,52,53,54,56,57], CO in 5 studies [39,42,56,57,58], LVEF in 12 studies [26,27,28,30,32,33,38,48,51,52,58,59] and LVM in 2 studies [40,54].

Sixteen studies measured acute [22,23,24,25,27,29,31,32,34,39,41,47,49,51,56,57] effect. In total, 4 included studies evaluated the acute and sub-acute effects of CCB [35,36,48,50], 5 evaluated only sub-acute effects [28,43,44,53,59] and 13 studies measured the chronic effects of CCB treatment [23,26,30,33,37,38,40,42,45,46,52,54,55,58].

Study designs consisted of 18 randomized controlled trials (RCT) [23,26,28,29,30,31,34,37,40,41,42,43,44,45,52,54,58,59] of which three were also a crossover study (34, 41, 52). Of the other studies, 1 was a retrospective cohort study [33] and 19 were prospective cohort studies [22,24,25,27,32,35,36,38,39,46,47,48,49,50,51,53,55,56,57]. Of the included studies containing CCB interventions, 27 studies included only male subjects [22,24,25,26,27,28,29,30,31,33,34,36,37,39,40,41,46,47,48,50,52,53,54,56,57,58,59], none included only female subjects, and the remaining 11 studies contained subjects of both sexes [23,32,35,38,42,43,44,45,49,51,55]. Publication bias assessed via Eggers’s regression showed significant bias for HR in males, but no significant bias for all other variables included (Table 2).

**Table 1 biomedicines-11-01622-t001:** Characteristics of studies.

Study	Patient	Ethnicity	CCB Treatment(Administration)	Mean Dose (mg/Day)	% Max Dose *	Subjects CCB (n)			Control Group **	Controls (n)			Age (Years + SD)	Intervention Duration(Days)	Study Design	Extracted Variables	Mentioned Method(s) of Measurement
						Total	M	F		Total	M	F					
Mehlum (2020) [45]	HTN, DM, HF, MI, LVH	W, B, A	Amlodipine (oral)	5	0.5	7477	4305	3172	Valsartan	7519	4332	3187	67.2 (8.1)	180	RCT	SBP, DBP, HR, MAP	Sphygmomanometry
Baysal (2017) [44]	HTN	-	Amlodipine (oral)	10	1	38	22	16	Telmisartan	39	22	17	48.0 (10)	30	RCT	SBP, DBP	Sphygmomanometry, ECG, echo
Thuc Sinh (2015) [43]	HTN, DM	-	Amlodipine (oral)	10	1	52	34	18	-	-	-	-	65.3 (10.7)	28	RCT	SBP, DBP,	Sphygmomanometry
			Lercanidipine (oral)	20	1	52	32	20									
Lindqvist (2007) [42]	HTN	-	Amlodipine (oral)	7.5	0.75	14	11	3	-	-	-	-	59 (***)	42	RCT	HR, CO	Catheterization, ECG, echo
			Mibefradil (oral)	75	0.75	14	11	3									
Petrella (2000) [53]	HF	-	Verapamil (oral)	240	0.33	10	10	0	-	-	-	-	73.0 (4.0)	26	Prospective cohort	HR	Echo
Burggraaf (1998) [34]	Healthy	-	Nifedipine (oral)	20	0.33	9	9	0	Captopril	9	9	-	18–35 (***)	0.125	RCT, crossover	HR, MAP	Echo
Gottdiener (1998) [40]	HTN	W, B	Diltiazem (oral)	240	0.67	185	185	0	Atenolol, captopril, clonidine, hydrochlorothiazide or prazosin	920	920	0	58.8 (10)	730	RCT	SBP, DBP, LVM, HR	Sphygmomanometry
Goldsmith (1997) [41]	HF	-	Amlodipine (oral)	7.5	0.75	7	7	0	Placebo	7	7	0	56 (***)	10	RCT, crossover	MAP, HR	Sphygmo-manometry, echo
Tomiyama (1997) [33]	HT	-	Nifedipine (oral)	30	0.5	13	13	0	Acebutolol	9	9	0	46 (7)	1095	Retrospective cohort	SBP, DBP, LVEF	Sphygmo-manometry, echo
Seki (1996) [32]	MI	-	Nifedipine (sublingual)	10	0.17	8	7	1	-	-	-	-	63 (10)	0.021	Prospective cohort	HR, LVEF	Catheterization
Naritomi (1995) [55]	HTN	-	Nitrendipine (oral)	10	0.25	10	7	3	-	-	-	-	60.5 (***)	56	Prospective cohort	SBP, DBP	Sphygmo-manometry
Risoe (1993) [31]	HF, MI	-	Nifedipine (sublingual)	20	0.33	8	4	0	Untreated	4	4	0	****	0.03	RCT	HR	Catheterization
Heywood (1991) [39]	HF, CAD	-	Diltiazem (iv)	25	0.07	9	9	0	-	-	-	-	68 (9)	0.02	Prospective cohort	SBP, DBP, CO, HR	Sphygmo-manometry, echo, ecg, catheterization
Sheiban (1991) [30]	HTN	-	Nifedipine (oral)	52	0.87	7	7	0	Untreated	10	10	0	41 (8.1)	180	RCT	LVEF	Echo, ecg, sphygmo-manometry
			Lacidipine (oral)	5	0.83	8	8	0									
Bekheit (1990) [29]	MI	-	Diltiazem (oral)	180	0.5	9	0	0	Metoprolol	8	8	0	62 (13)	6	RCT	SBP, DBP, HR	Ecg, sphygomomanometry
			Nifedipine (oral)	30	0.5	10	0	0									
Senda (1990) [38]	HTN, LVH	-	Diltiazem (oral)	180	0.5	9	6	3	-	-	-	-	60 (***)	180	Prospective cohort	LVEF	Echo, ecg, sphygomomanomatry
Setaro (1990) [52]	HF, HTN, CAD, MI, DM	W, B	Verapamil (orally)	256	0.36	20	20	0	Placebo (4 day washout interval after verapamil)	20	20	0	68 (5)	32	RCT, crossover	SBP, DBP, HR, LVEF	Echo, ECG, sphygomomanometry
Binetti (1989) [57]	HF, CAD,	-	Felodipine (iv)	0.85	0.85	10	10	0	-	-	-	-	53 (***)	0.042	Prospective cohort	SBP, DBP, CO, MAP, HR	Catheterization, ecg, sphygomomanometry
Crawford (1989) [28]	HF	-	Nifedipine (oral)	66	1.1	10	10	0	Digoxin, hydralazin (same patients, after CCB)	10	10	0	54 (***)	30	RCT	SBP, LVEF, HR	Ecg,
La Rovere (1989) [50]	MI	-	Nicardipine (iv and oral)	5 (iv)	0.01 (iv)	10	10	0	-	-	-	-	54 (9)	0.0069 (iv)	Prospective cohort	HR	Ecg,
				60 (oral)	0.17 (oral)									21 (oral)			
McGrath (1989) [58]	HF	-	Isradipine(oral)	15	0.75	9	9	0	Placebo	9	9	0	54 (***)	84	RCT	CO, LVEF, HR	Catheterization
Szlachcic (1989) [37]	HTN	-	Diltiazem (oral)	240	0.67	13	13	0	Placebo	11	11	0	48 (10)	112	RCT	SBP, DBP	Echo, sphygomomanometry, ecg
Bostr.m (1988) [36]	MI	-	Diltiazem (oral)	120	0.33	12	12	0	-	-	-	-	61 (***)	0.104	Prospective cohort	SBP, HR	Ecg, sphygomomanometry
				180	0.5									14			
Fisman (1988) [59]	MI, CAD	-	Gallopamil (oral)	75	0.38	9	9	0	Placebo	6	6	0	60.3 (5.5)	21	RCT	LVEF	Echo
				112.5	0.56	9	9	0									
				150	0.75	8	8	0									
Mookherjee (1988) [27]	HF	-	Nifedipine (sublingual)	80	1.33	12	12	0	-	-	-	-	55–77 (***)	1	Prospective cohort	LVEF, HR	Echo, catheterization
Burlew (1987) [49]	HF, CAD, HTN	-	Nicardipine (oral)	75	0.21	10	7	3	-	-	-	-	54 (***)	9	Prospective cohort	HR	Catheterization, ecg
Fagard (1987) [56]	HTN	-	Felodipine (oral)	6	0.6	10	10	0	-	-	-	-	41 (9)	0.06	Prospective cohort	HR, CO	Echo, catheterization
Giles (1987) [54]	HTN, LVH	W, B	Nitrendipine (oral)	20	0.5	9	9	0	Hydrochlorothiazide	9	0	0	66 (3)	56	RCT	SBP, DBP, HR, LVM,	Echo, sphygomomanometry
Sheiban (1987) [26]	HTN	-	Nifedipine (oral)	30	0.5	8	8	0	Captopril	8	8	0	38 (10)	180	RCT	SBP, DBP, LVEF, HR	Echo, sphygomomanometry
Lahiri (1986) [48]	HF, MI	-	Nicardipine (oral)	10	0.03	10	10	0	-	-	-	-	63 (***)	0.02	Prospective cohort	LVEF	Sphygomomanometry
				90	0.25									28			
Ortiz (1986) [51]	HTN	-	Verapamil (oral)	240	0.33	18	5	13	-	-	-	-	53.6 (***)	0.125	Prospective cohort	SBP, DBP, HR, LVEF	Echo, auscultation method
Kubo (1985) [24]	HF	-	Nifedipine (oral)	10	0.17	7	7	0	-	-	-	-	60 (***)	0.08	Prospective cohort	HR	Catheterization,
Nakamura (1985) [25]	HF	-	Nifedipine (sublingual)	20	0.33	8	8	0	-	-	-	-	55 (***)	0.02	Prospective cohort	HR	Echo, ecg
Silke (1984) [47]	CAD	-	Nicardipine (iv)	1.25	0.003	10	10	0	-	-	-	-	47 (***)	0.08	Prospective cohort	HR	Ecg
				2.5	0.007												
				5	0.01												
				10	0.03												
Suwa (1984) [35]	LVH, HF	-	Diltiazem (iv and oral)	10 (iv)	0.03	13	11	2	Propranol	13	11	2	43 (***)	0.02 (iv)	Prospective cohort	SBP, DBP, HR	Ecg, echo, sphygomomanometry
				180 (oral)	0.5									14 (oral)			
Amende (1983) [22]	CAD	-	Nifedipine (iv)	0.1 (iv)	0.002	8	8	0	-	-	-	-	53.5 (***)	0.007	Prospective cohort	HR	Ecg
Fujita (1983) [46]	HTN	-	Nicardipine (oral)	60	0.17	10	10	0	-	-	-	-	52.1 (1.7)	120	Prospective cohort	HR	Echo, sphygomomanometry
Paulus (1983) [23]	LVH, HF	-	Nifedipine (sublingual)	10	0.17	10	3	7	Nitroprusside	10	3	7	48.3 (***)	0.02	RCT	HR	Echo, ecg

Data presented as mean ± SD or percentages. DM = diabetes mellitus, HTN = hypertension, HF = heart failure, CAD = coronary artery disease, MI = myocardial infarction, LVH = left ventricular hypertrophy, W = white, B = black, A = Asian, SD = standard deviation, RCT = randomized controlled trial, MAP = mean arterial pressure, SBP = systolic blood pressure, DBP = diastolic blood pressure, HR = heart rate, LVEF = left ventricular ejection fraction, LVM = left ventricular mass, ECG = electrocardiography, echo = echocardiography. * Percentage of maximal dosage for the indication hypertension. Amlodipine 10 mg/day orally [60]; Lercanidipine 20 mg/day [61]; Mibefradil 100 mg/day [62]; Nicardipine 360 mg/24 h [63]; Verapamil 720 mg/day orally [64]; Nifedipine 60 mg/day [65]; Diltiazem 360 mg/day [66]; Nitrendipine 40 mg/day [67]; Lacidipine 6 mg/day [68]; Felodipine 1 mg/24 h (iv) 10 mg/day (orally) [69]; Isradipine 20 mg/day [70]; Gallopamil 200 mg/day [71]. ** Control group: other antihypertensive treatment, placebo or non-drug intervention. *** SD not reported. **** Not mentioned.

### 3.3. Quality Assessment

Table 3 summarizes the quality assessment per domain according to the RoB2 tool [16]. Six studies had a low overall risk of bias [34,40,52,54,58,59]. Twenty-eight studies were rated with a high overall risk of bias [22,23,24,25,27,28,29,30,31,32,35,36,38,39,41,43,44,45,47,48,49,50,51,53,55,56,57,59]. The remaining four studies were scored as having some concerns [26,33,37,46].

### 3.4. Systolic Blood Pressure

The mean difference and relative change from baseline in percentage for systolic blood pressure (SBP) are reported in Table 4 and Figure 2. The mean SBP in the female population was 162.6 mmHg (95% CI 156.9; 168.2) and the mean SBP in the male population was 146.0 mmHg (95% CI 140.1; 151.8) (*p* < 0.0001). In females as compared to males, SBP decreased by −27.6 mmHg (95% CI −36.4; −18.8) (%change −17.1 (95% CI −22.5; −11.6)) versus a decrease of 14.4 mmHg (95% CI −19.0; −9.9) (%change, −9.8 (95% CI −12.9; −6.7)). This change was statistically significant between sexes (*p*-value = 0.009). Heterogeneity was moderate to high in female (I^2^ = 77%) and high in male (I^2^ = 83%) data. Heterogeneity in SBP response was significantly affected by the CCB diltiazem, delineated as a clinical source of heterogeneity (Table 5).

The mean difference for SBP by treatment duration is reported in Table 6. In females, the observed decrease in SBP is greatest in the acute and sub-acute treatment phase, while in males the observed decrease is largest during sub-acute and chronic treatment (Figure 3, Figure 4 and Figure 5).

### 3.5. Diastolic Blood Pressure

The mean DBP in the female population was 93.1 mmHg (95% CI 88.0; 98.2) and the mean DBP in the male population was 93.2 mmHg (95% CI 89.1; 97.3) (*p*-value = 0.973). DBP decreased by −14.1 (95% CI −18.8; −9.3) (%change, −15.2 (95% CI −20.3; −10.1)) in females as compared to −10.6 mmHg (95% CI −14.0; −7.3) (%change −11.2 (95% CI −14.8; −7.7)) in males (Table 4, Figure 6). This effect did not reach statistical significance between sexes (*p*-value = 0.244). Heterogeneity was moderate to high in female (I^2^ = 61%) and high in male (I^2^ = 94 %) data. Only one clinical source of heterogeneity (diltiazem) significantly affected the change in DBP (Table 5). The mean difference for DBP by treatment duration is reported in Table 6. The observed decrease in DBP is, in females, the largest after acute and sub-acute treatment. For males, the effect is the greatest in the sub-acute phase (Figure 7, Figure 8 and Figure 9).

### 3.6. Mean Arterial Pressure

The mean MAP in the female population was 110.6 mmHg (95% CI 110.2; 111.0) and the mean MAP in the male population was 95.6 mmHg (95% CI 82.7; 108.5) (*p*-value = 0.023). In females, MAP changed by −13.9 mmHg (95% CI −14.5; −13.2) (%change −12.5 (95% CI −13.1; −12.0)) as compared to −8.7 mmHg (95% CI −14.1; −3.3) (%change, −8.9 (95% CI −14.5; −3.4)) in males (Table 4, Figure 10) but the difference between sexes (*p*-value = 0.061) did not reach statistical significance. Heterogeneity could not be calculated in females as only one study was included. Heterogeneity was high (I^2^ = 85%) in males. The clinical source of heterogeneity detected by meta-regression analysis was nifedipine. Methodological sources of heterogeneity did not significantly contribute to the observed change in MAP (Table 5).

### 3.7. Heart Rate

The mean HR in the female population was 74.8 bpm (95% CI 68.0; 81.7) and the mean HR in the male population was 74.4 bpm (95% CI 72.9; 76.0) (*p*-value = 0.919). Heart rate (beats per minute (bpm)) decreases in females (−1.8 bpm (95% CI −2.5; −1.2) (%change −2.5 (95% CI −3.4; −1.6)). In males, HR did not change appreciably (0.3 bpm (95% CI −1.2; 1.8) (%change 0.4 (95% CI −1.7; 2.4)) (Table 4, Figure 11). This sex-difference is significant (*p*-value = 0.011). The heterogeneity is low in females I^2^= 0% and moderate in males I^2^ = 72%. Three clinical sources of heterogeneity (felodipine, mibefradil and nifedipine) significantly affected the change in HR (Table 5).

### 3.8. Cardiac Output

The mean CO in females was 5.0 L/min (95% CI 3.6; 6.5) and in males 4.7 L/min (95% CI 4.2; 5.1) (*p*-value = 0.635). A comparable effect between females and males was observed in CO after CCB use, which was also not statistically significant between sexes (*p*-value = 0.244). CO remained unaltered in females (−0.2 L/min (95% −1.8; 1.4) (%change −4.0 (95% CI −35.8; 27.7)) but significantly increased in males by 0.8 L/min (95% CI 0.1; 1.6) (%change 18.2 (95% CI 2.1; 34.2)) (Table 4, Figure 12). Heterogeneity is low in female data (I^2^ = 0%), as one study could be included, and moderate to high in male data (I^2^ = 71%). The clinical source of heterogeneity detected by meta-regression analysis was felodipine and mibefradil. The moderate study quality and treatment duration, both methodological sources of heterogeneity, did also significantly contribute to the observed change in CO (Table 5).

### 3.9. Left Ventricular Ejection Fraction

The mean LVEF in the female population was 68.7 % (95% CI 65.9; 71.5) and the mean LVEF in the male population was 51.2% (95% CI 42.0; 60.4) (*p*-value = 0.0004). LVEF change was opposite in females as compared to males (Table 4, Figure 13). In females, a decrease in LVEF by −7.8 % (95% CI −12.4; −3.2) (%change −11.4 (95% CI −18.0; −4.7) was observed, whereas in males, LVEF remained unaltered (2.9% (95% CI −0.4; 6.1) (%change 5.3 (95% CI −0.7; 11.3)) (Table 3). The sex-difference was statistically significant (*p*-value < 0.001). Heterogeneity is low in female data (I^2^ = 0%) and moderate to high in male data (I^2^ = 77%). The change in LVEF was not significantly affected by clinical or methodological sources of heterogeneity (Table 5).

### 3.10. Left Ventricular Mass

LVM could only be extracted from two studies, and only from males. The mean LVM in the population was 322.5 g (95% CI 304.2; 340.8). In these studies, LVM remained unaltered by CCB (−15.9 g (95% CI −48.2; 16.4) (%change −4.9 (95% CI −15.0; 5.1))) (Table 4, Figure 14). Heterogeneity was low in these studies (I^2^ = 0%). 

## 4. Discussion

In this systematic review and meta-analysis, we observed that CCB treatment significantly lowered SBP, DBP and MAP in both sexes but that the decrease is greater in females as compared to males. HR and LVEF decreased in females, while remaining unaltered in males. In females, prolonged treatment duration slightly attenuated the blood pressure-lowering effect, whereas in males the effect on blood pressure was the highest after 14 days of treatment. 

System-biological cardiovascular and regulatory differences between sexes are likely underlying possible differences in responsiveness to CCB in the treatment of hypertension [13,72,73]. These differences may be expected in the most important blood pressure regulatory systems amongst sympathetic nervous system (SNS), renin-angiotensin system [29], endothelin-1 (ET-1), vasomotion and sex hormones [73].

The SNS regulates the vascular tone by mediating vasoconstriction and vasodilation by transmitting norepinephrine targeting alpha- and beta adrenoceptors [74,75]. Studies have shown that sensitivity to adrenergic stimulation is sex-specific, where males have greater sensitivity to norepinephrine compared to females, leading to more vasoconstriction in males [76,77,78,79]. These findings may be affected by the presence of estrogen receptors on endothelial and vascular smooth muscle cells. Stimulation by estrogen results in weaker vascular reactivity to adrenergic stimulation [79].

The RAS consists of a classic and non-classical pathway. The classic pathway is currently defined as the ACE-Ang II AT1R axis promoting vasoconstriction, sodium, water retention as well as inflammation, oxidative stress, cellular growth, and fibrosis. On the contrary, the non-classical RAS pathway is composed primarily of the angiotensin-(1-7)-ACE2-MasR/AT2R pathways which opposes the actions of the classical stimulated Ang II-AT1R axis by causing an increase in nitric oxide and prostaglandins, mediating vasodilation, natriuresis, diuresis, and lowering oxidative stress. Female estrogen mediates downregulation of angiotensin II and upregulation of angiotensin-(1-7)-ACE2-MasR/AT2R pathways causing vasodilatation, whereas male testosterone stimulates the classic pathway leading to vasoconstriction [80,81,82,83,84,85,86]. Another important vasoconstrictor is ET-1, which is secreted by endothelial cells. Sex hormones influence the release of ET-1 in opposite ways, whereby testosterone causes an increase in ET-1 release and estrogen and progesterone cause inhibition of ET-1 release [73,87].

The prevalence of hypertension increases with age in both sexes. Remarkable is the higher incidence of hypertension in younger males compared to younger females. However, as age increases, more females suffer from hypertension compared to age-matched men. This may depend on menopause causing a drop in estrogen [88,89]. Clinical studies suggest that estrogen protects against hypertension by stimulation of the vasodilator pathway mediated by nitric oxide and prostacyclin and inhibition of the vasoconstrictor pathway mediated by the sympathetic nervous system and angiotensin [90,91].

Pharmacologically, CCB act by binding to the L-type long-acting voltage gated-calcium channels in the heart, vascular smooth muscles (coronary and peripheral arterials) and pancreas. Blocking the calcium channels leads to inhibition of Ca^2+^ influx into excitable cells preventing Ca^2+^ to serve as an intracellular messenger. As intracellular Ca^2+^ leads to vasoconstriction, CCB leads to vasodilatation. 

Two types of CCBs are known. First are the non-dihydropyridines, amongst verapamil, primarily influencing the sinoatrial and atrioventricular node resulting in a reduction in cardiac conduction and contractility. Therefore, this group is mostly used for treating hypertension, reduction in oxygen demand and controlling heart rate in arrhythmias. Verapamil is known to be one of the first well characterized P-glycoprotein (P-gp) substrates, as well as a substrate of CYP3A4 resulting in extensive first-pass hepatic and intestinal metabolism [13]. The second group is dihydropyridines, for example, amlodipine. Dihydropyridines are known for their vasodilating effect on peripheral arterials, useful in hypertension treatment, migraines and cranial vasospasms [12,14]. Amlodipine is not generally considered to be a P-gp substrate and has low rates of first-pass metabolism and a high bioavailability [13].

Splitting the analysis into a non-dihydropyridines and a dihydropyridines group results in comparable findings according for BP. However, HR seems to decrease more in males compared to females using non-dihydropyridines. LVEF, CO and LVM are hard to compare as only few included studies investigated those parameters (Appendix A).

Clinically, previous studies comparing the effects of CCB between females and males show sex differences for some types of CCB, which might relate to differences in metabolization and metabolic rate. In a study on the influence of sex on the pharmacokinetics of verapamil and norverapamil, Dadashzadeh et al., showed the blood concentration of norverapamil compared to verapamil to be significantly higher in females [92]. The substrate CYP3A4 metabolizes verapamil to norverapamil. As the mean residence time of norverapamil was significantly shorter in females compared to males, the study concluded that the production of norverapamil is more extensive in females and therefore a sex-dependent process, most likely because of a higher activity of CYP3A4 or lower activity of P-gp leading to faster clearance. Higher expression of CYP3A4 in females has been found in previous studies [13,93,94,95]. However, some studies showed the contrary; Cummins at al. suggest that sex-specific differences depend on P-gp activity [96]. If a drug is not only metabolized by CYP3A4 but is also a substrate of P-gp, intracellular hepatic levels will rise, resulting in more opportunities to encounter its metabolizing enzymes and higher clearance rates, even when enzyme protein levels are equal in sexes. Cummins concludes that intracellular drugs concentration will not differ between sexes if a drug is a substrate of CYP3A4 but not a P-gp substrate. 

Several studies have assessed the efficacy and safety of amlodipine which has been safely used in patients with New York Heart Association classes II and III heart failure. Kloner et al. conducted a prospective, multicentric trial including 1084 patients (mean age 55.5 years; 35% females and 65% males) with mild to moderate hypertension and showed greater DBP changes from baseline in females (91.4%) compared to males (83.0%) (*p*-value = 0.001) using amlodipine [97]. These results remained significant after adjusting for baseline differences in, e.g., weight, age and weight normalized dose [97,98]. The authors suggest that this difference in response can be explained by vascular reactivity, distribution, metabolism of the drug and differences in the etiology of hypertension between females and males [13]. Kang et al. conducted a study on the effect of age on the oral clearance of amlodipine [99]. Approximately 210 elderly subjects (mean age females 79 years, males 72 years) were included and showed significant effects for sex. In accordance with previous research on other substrates of CYP3A4, faster clearance in females compared to males was observed. Contributing to the hypothesis that CYP3A4 expression in female liver tissue is greater than in males are many in vivo studies showing a greater decrease in CYP3A4 in aging males compared to aging females [100]. This should be taken into consideration as older patients are most likely to receive antihypertensive medication.

### Strengths and Limitations

One of the strengths of our systematic review and meta-analysis is the large number of studies we examined for inclusion. This extensive search resulted in a population of 8202 patients using CCB and having sex-stratified data available. Another strength is that two independent reviewers screened all articles. Furthermore, we used the RoB2-tool, as recommended by Cochrane, to detect forms of bias.

On the other hand, some limitations should be mentioned. First, our meta-analysis included remarkably more male patients compared to female patients (60.2% vs. 39.8%), causing female underrepresentation. However, this is rather a clinical reflection than a lack of our study selection method. Second, the ROB2 tool classified more than half of the included articles as ‘high risk of bias’. This could be explained as the ROB2 tool classifies non-RCTs lower than RCTs. Half of the included articles were prospective non-RCTs and therefore qualified as lower quality, contributing to the high risk of bias. This means that we most likely overestimated the risk of bias for some studies. Third, clinical sources of heterogeneity (use of different CCB) significantly affected the change for some parameters. Treatment duration, a methodological source of heterogeneity, significantly contributed to the observed change in CO. Furthermore, in a lot of the included studies patients received co-medication during the study protocol. On one hand, this could have biased the observed effect. On the other hand, since co-medication was used before and after CCB initiation, this bias could be considered constant. Maybe more importantly, this could be a reliable representation of reality, contributing to the external validity of our study. 

## 5. Conclusions and Recommendations

In individuals with hypertension, CCB lowers BP in both sexes. However, the decrease, especially in SBP, is significantly greater in females as compared to males. The lowering effect on HR and LVEF is only significant in females. When normalizing blood pressure, differences in these sex-dependent effects may be taken into account. 

## Figures and Tables

**Figure 1 biomedicines-11-01622-f001:**
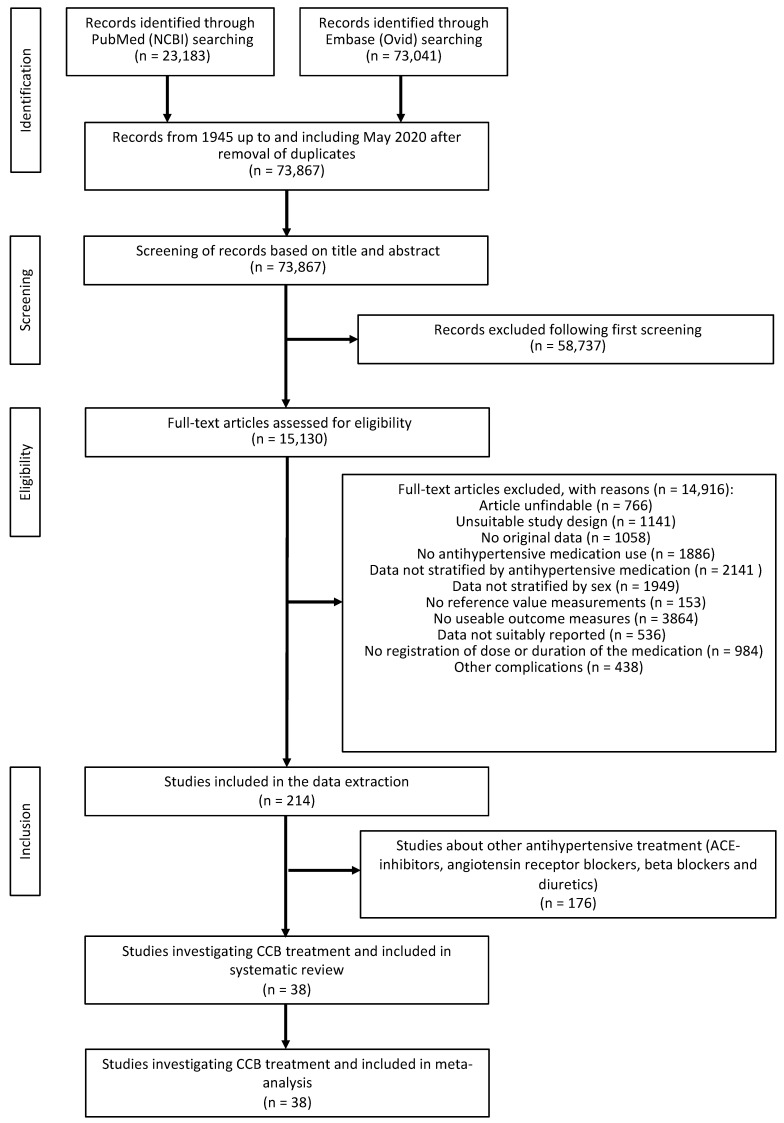
Flowchart of systematic selection process.

**Figure 2 biomedicines-11-01622-f002:**
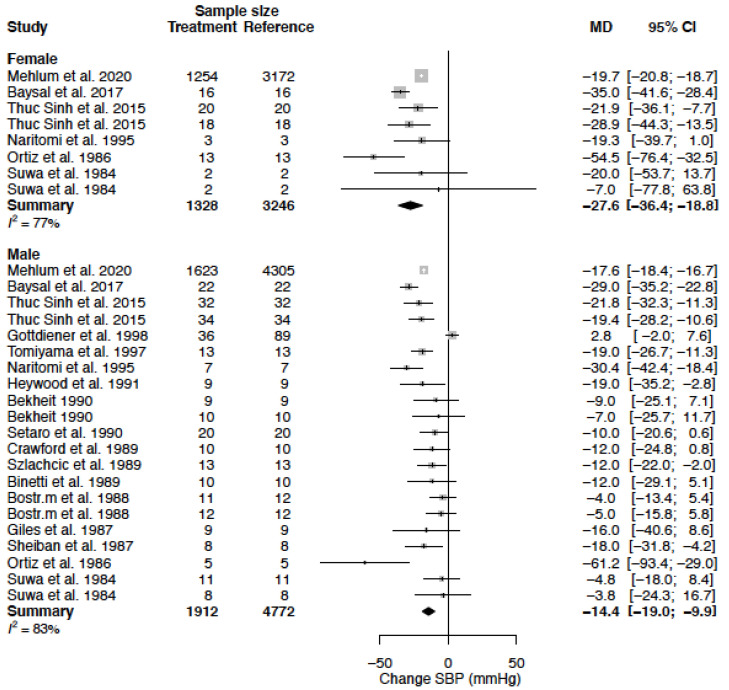
Forest plot of systolic blood pressure (SBP) change in mmHg after CCB use compared to baseline for females and males [26,28,29,33,35,36,37,39,40,43,44,45,51,52,54,55,57]. MD = mean difference.

**Figure 3 biomedicines-11-01622-f003:**
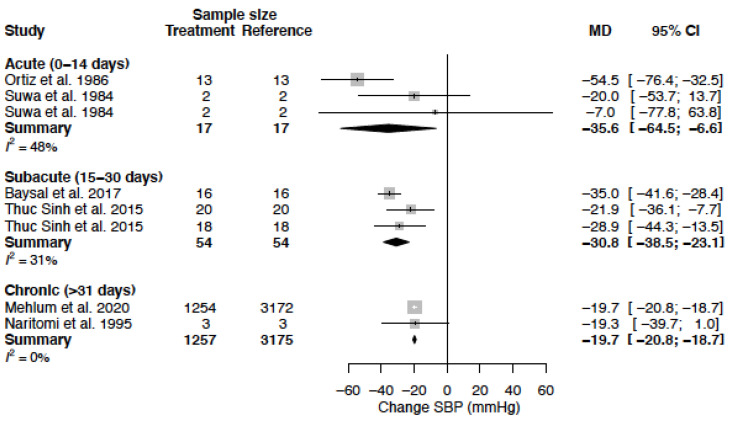
Forest plot of systolic blood pressure (SBP) change in mmHg after acute, sub-acute and chronic CCB use compared to baseline for females [35,43,44,45,51,55]. MD = mean difference.

**Figure 4 biomedicines-11-01622-f004:**
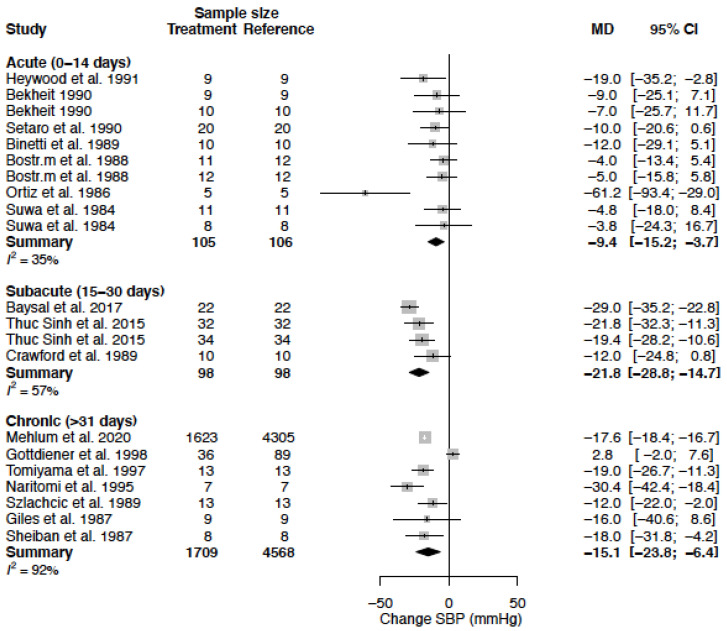
Forest plot of systolic blood pressure (SBP) change in mmHg after acute, sub-acute and chronic CCB use compared to baseline for males [26,28,29,33,35,36,37,39,40,43,44,45,51,52,54,55,57]. MD = mean difference.

**Figure 5 biomedicines-11-01622-f005:**
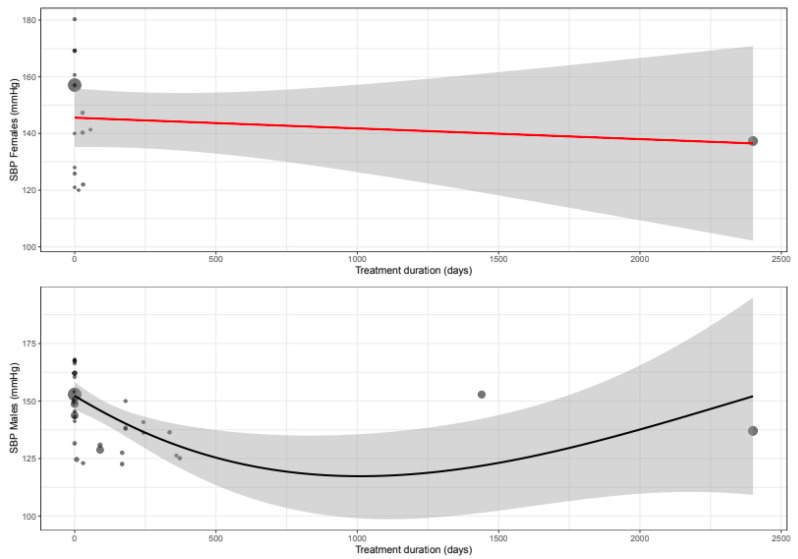
Meta-regression curve of systolic blood pressure (SBP) by treatment duration (days). Every circle represents one article and the size represents the number of participants included in the study, shown as a small or larger circle.

**Figure 6 biomedicines-11-01622-f006:**
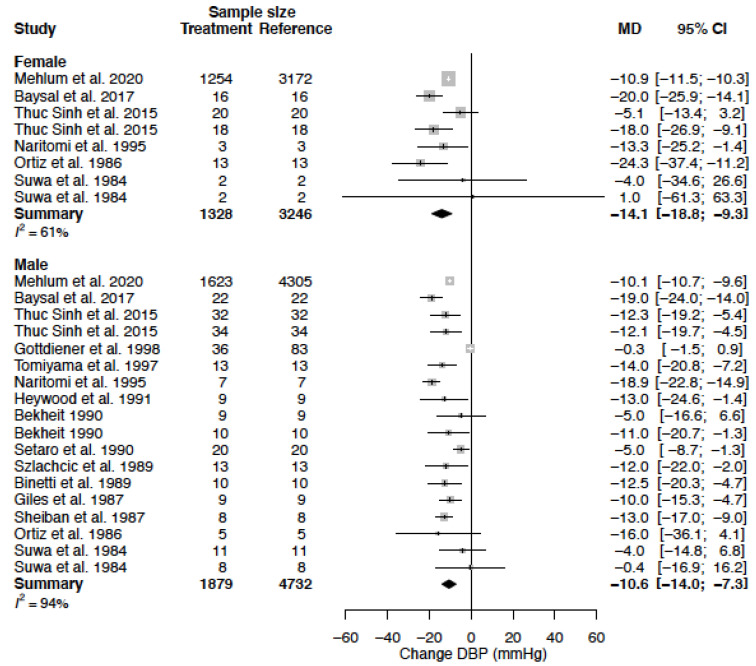
Forest plot of diastolic blood pressure (DBP) change in mmHg after CCB use compared to baseline for females and males [26,29,33,35,37,39,40,43,44,45,51,52,54,55,57]. MD = mean difference.

**Figure 7 biomedicines-11-01622-f007:**
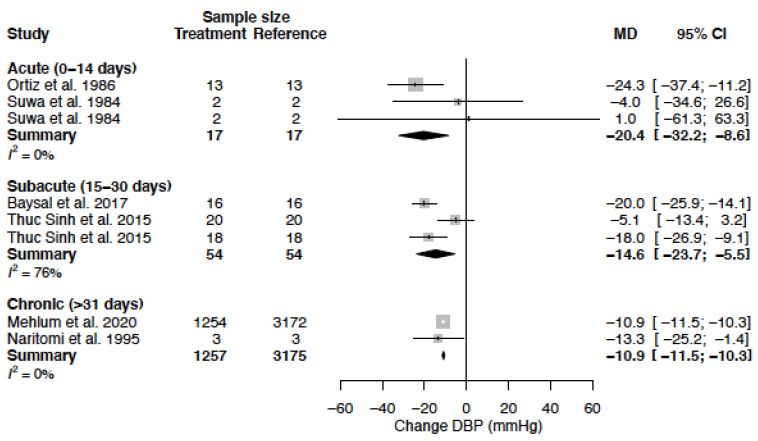
Forest plot of diastolic blood pressure (DBP) change in mmHg after acute, sub-acute and chronic CCB use compared to baseline for females [35,43,44,45,51,55]. MD = mean difference.

**Figure 8 biomedicines-11-01622-f008:**
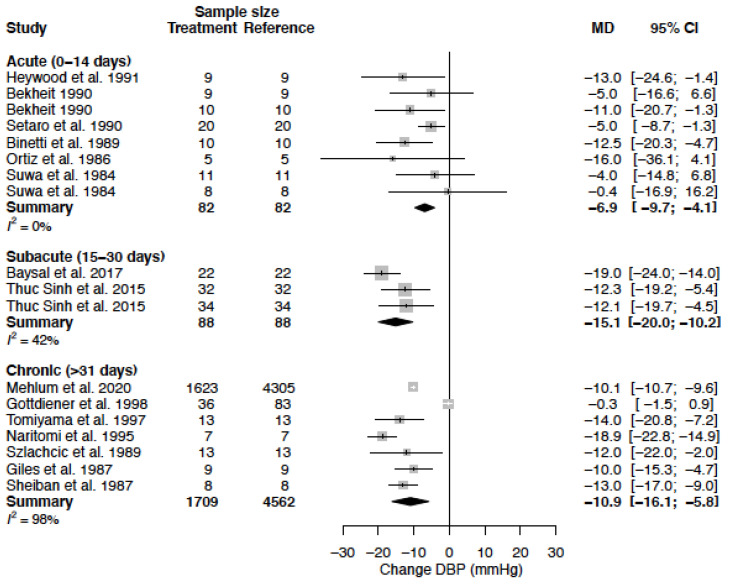
Forest plot of diastolic blood pressure (DBP) change in mmHg after acute, sub-acute and chronic CCB use compared to baseline for males [26,29,33,35,37,39,40,43,44,45,51,52,54,55,57]. MD = mean difference.

**Figure 9 biomedicines-11-01622-f009:**
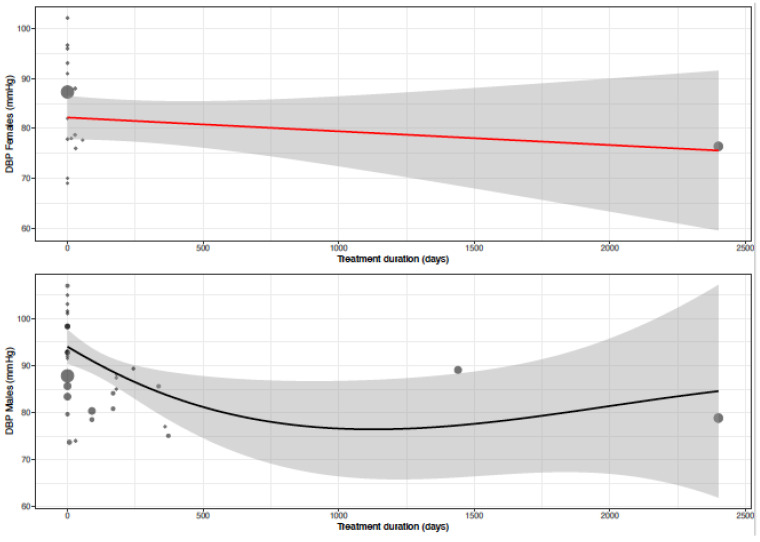
Meta-regression curve of diastolic blood pressure (DBP) by treatment duration (days). Every circle represents one article and the size represents the number of participants included in the study, shown as a small or larger circle.

**Figure 10 biomedicines-11-01622-f010:**
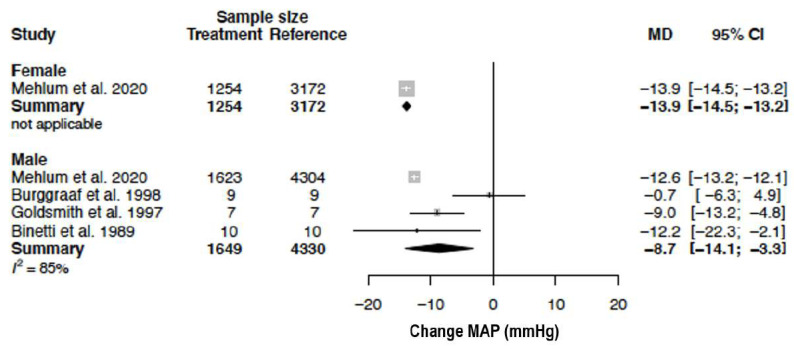
Forest plot of mean arterial pressure (MAP) change in mmHg after CCB use compared to baseline for females and males [34,41,45,57]. MD = mean difference.

**Figure 11 biomedicines-11-01622-f011:**
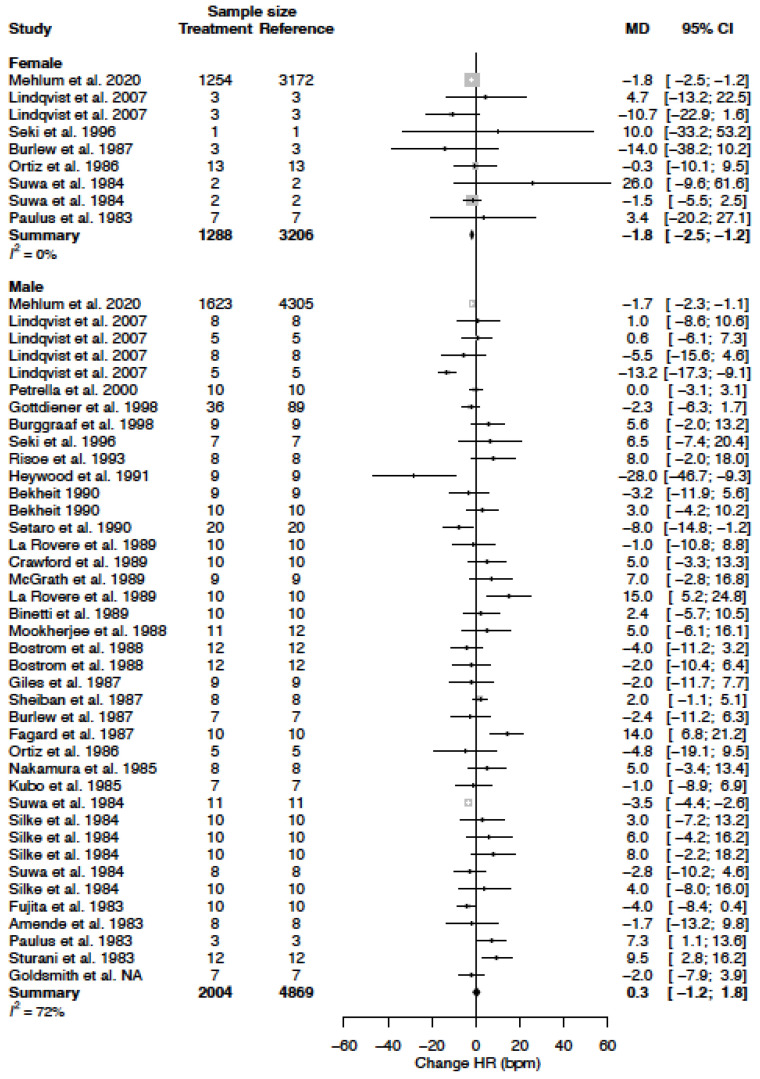
Forest plot of heart rate (HR) change in bmp after CCB use compared to baseline for females and males [22,23,24,25,26,27,28,29,31,32,34,35,36,39,40,41,42,45,46,47,49,50,51,52,53,54,56,57]. MD = mean difference.

**Figure 12 biomedicines-11-01622-f012:**
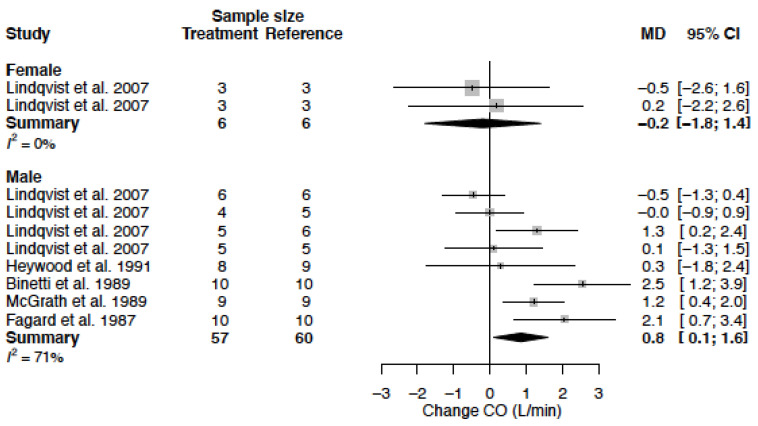
Forest plot of cardiac output (CO) change in L/min after CCB use compared to baseline for males and females [39,42,56,57,58]. MD = mean difference.

**Figure 13 biomedicines-11-01622-f013:**
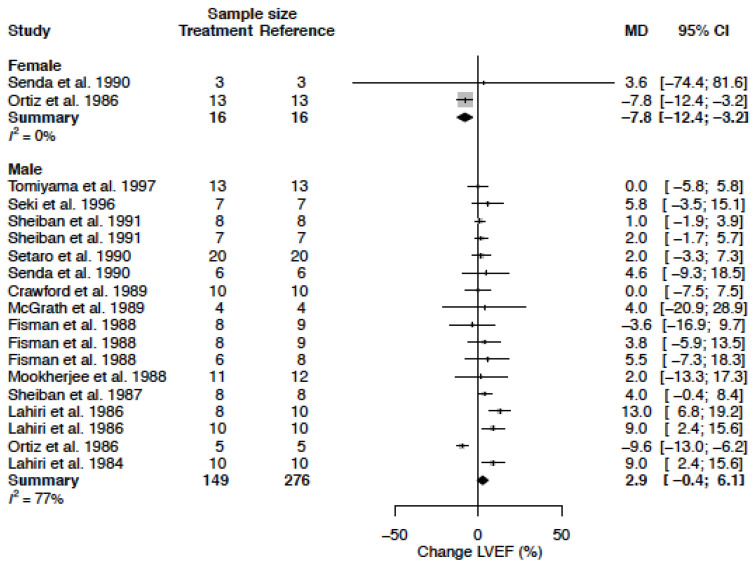
Forest plot of left ventricular ejection fraction (LVEF) change in % after CCB use compared to baseline for females and males [26,27,28,30,32,33,38,48,51,52,58,59]. MD = mean difference.

**Figure 14 biomedicines-11-01622-f014:**
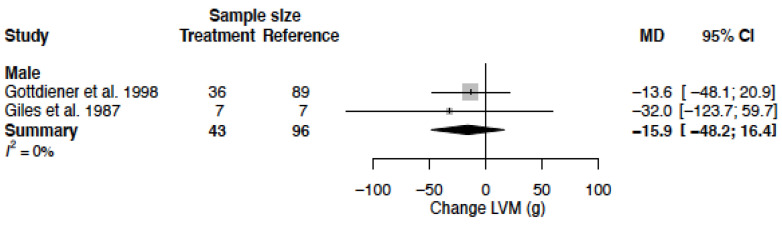
Forest plot of left ventricular mass (LVM) change in grams after CCB use compared to baseline for males [40,54]. MD = mean difference.

**Table 2 biomedicines-11-01622-t002:** Publication bias using Eggers’s regression for all variables.

	Male	Female	cMD Male	cMD Female
SBP	0.3503	0.1627	-	-
DBP	0.7937	0.3185	-	-
MAP	0.2550	-	-	-
HR	0.0283	0.6273	−2.09 (−3.60; −0.58)	-
CO	0.4162	-	-	-
LVEF	0.1651	-	-	-
LVM	-	-	-	-

**Table 3 biomedicines-11-01622-t003:** Quality assessment.

	Random Sequence Allocation (Selection Bias)	Allocation Concealment (Selection Bias)	Incomplete Outcome Data (Attrition Bias)	Measure-ments Outcomes (Detection Bias)	Selective Reporting (Reporting Bias)	Overall Bias
Lindqvist et al. (2007) [42]	Low	High	High	Low	High	High
McGrath et al. (1989) [58]	Low	Low	Low	Low	Low	Low
Heywood et al. (1991) [39]	High	Some concerns	Low	Some concerns	Low	High
Binetti et al. (1989) [57]	High	Some concerns	Low	Low	Low	High
Fagard et al. (1987) [56]	High	Some concerns	Low	Low	Low	High
Mehlum et al. (2020) [45]	High	Low	Low	Low	Low	High
Baysal et al. (2017) [44]	Low	High	Low	High	High	High
Thuc Sinh et al. (2015) [43]	High	Some concerns	Low	Some concerns	Low	High
Gottdiener et al. (1998) [40]	Low	Low	Low	Low	Low	Low
Tomiyama et al. (1997) [33]	Some concerns	Low	Low	Low	Low	Some concerns
Naritomi et al. (1995) [55]	High	Low	Low	Low	Low	High
Setaro et al. (1990) [52]	Low	Low	Low	Low	Low	Low
Bekheit et al. (1990) [29]	High	Low	Low	Low	Low	High
Szlachcic et al. (1989) [37]	Low	Low	Low	Low	Some concerns	Some concerns
Sheiban et al. (1987) [26]	Low	Some concerns	Low	Low	Low	Some concerns
Giles et al. (1987) [54]	Low	Low	Low	Low	Low	Low
Ortiz et al. (1986) [51]	High	Low	Low	Low	Low	High
Suwa et al. (1984) [35]	High	Some concerns	Low	Low	Low	High
Seki et al. (1996) [32]	High	Low	Low	Low	Low	High
Sheiban et al. (1991) [30]	High	Low	Low	Low	Low	High
Senda et al. (1990) [38]	High	Low	Low	Low	Low	High
Crawford et al. (1989) [28]	High	Some concerns	Low	Low	High	High
Fisman et al. (1988) [59]	Low	Low	Low	Low	Low	Low
Mookherjee et al. (1988) [27]	High	Low	Low	Low	Low	High
Lahiri et al. (1986) [48]	High	Low	Low	Low	Low	High
Petrella et al. (2000) [53]	High	Low	Low	Low	Low	High
Burggraaf et al. (1998) [34]	Low	Low	Low	Low	Low	Low
Risoe et al. (1993) [31]	High	Low	Low	Low	Low	High
La Rovere et al. (1989) [50]	High	Some concerns	Low	Low	Low	High
Bostrom et al. (1988) [36]	High	Some concerns	Low	Low	Low	High
Burlew et al. (1987) [49]	High	Low	Low	Low	Some concerns	High
Kubo et al. (1985) [24]	High	Some concerns	Low	Low	Some concerns	High
Silke et al. (1984) [47]	High	Low	Low	Low	Low	High
Fujita et al. (1983) [46]	Some concerns	Some concerns	Low	Low	Low	Some concerns
Amende et al. (1983) [22]	High	Some concerns	Low	Low	Low	High
Paulus et al. (1983) [23]	High	Some concerns	Low	Low	Low	High
Goldsmith et al. (1997) [41]	High	Low	Low	Low	Low	High
Nakamura et al. (1985) [25]	High	Some concerns	Low	Low	Low	High

**Table 4 biomedicines-11-01622-t004:** Pooled changes in cardiovascular and haemodynamic parameters for females and males.

Parameter		Females	Males
SBP (mmHg)	MD %	−27.6 (−36.4; −18.8) −17.1 (−22.5; −11.6)	−14.4 (−19.0; −9.9) −9.8 (−12.9; −6.7)
DBP (mmHg)	MD %	−14.1 (−18.8; −9.3) −15.2 (−20.3; −10.1)	−10.6 (−14.0; −7.3) −11.2 (−14.8; −7.7)
MAP (mmHg)	MD %	−13.9 (−14.5; −13.2) −12.5 (−13.1; −12.0)	−8.7 (−14.1; −3.3) −8.9 (−14.5; −3.4)
HR (bpm)	MD %	−1.8 (−2.5; −1.2) −2.5 (−3.4; −1.6)	0.3 (−1.2; 1.8) 0.4 (−1.7; 2.4)
CO (L/min)	MD %	−0.2 (−1.8; 1.4) −4.0 (−35.8; 27.7)	0.8 (0.1; 1.6) 18.2 (2.1; 34.2)
LVEF (%)	MD %	−7.8 (−12.4; −3.2) −11.4 (−18.0; −4.7)	2.9 (−0.4; 6.1) 5.3 (−0.7; 11.3)
LVM (g)	MD %	- -	−15.9 (−48.2; 16.4) −4.9 (−15.0; 5.1)

Values are reported as mean difference (MD) and relative change (%) compared to baseline with 95% CI. SBP = systolic blood pressure, DBP = diastolic blood pressure, MAP = mean arterial pressure, HR = heart rate, CO = cardiac output, LVEF = left ventricular ejection fraction, LVM = left ventricular mass.

**Table 5 biomedicines-11-01622-t005:** *p*-values of meta-regression analysis.

Sources of Heterogeneity	SBP	DBP	MAP	HR	CO	LVEF
Diltiazem	0.0033	0.0169	-	0.3344	0.6689	-
Felodipine	0.4463	0.8744	0.8968	0.0013	0.0179	-
Isradipine	-	-	-	0.1107	0.4874	0.9644
Gallopamil	-	-	-	-	-	0.7769
Mibefradil	-	-	-	<0.0001	0.0469	-
Lacidipine	-	-	-	-	-	0.6467
Nicardipine	-	-	-	0.1601	-	0.4624
Lercanidipine	0.7665	0.2490	-	-	-	-
Nifedipine	0.3323	0.8709	0.0059	0.0005	-	0.7478
Nitrendipine	0.8774	0.8796	-	0.9458	-	-
Verapamil	0.4007	0.4838	-	0.8234	-	0.2200
Low quality	0.0765	0.0954	0.8467	0.8485	-	0.8701
Moderate quality	0.2036	0.8437	0.8034	0.3195	0.0015	0.8598
Treatment duration	0.5907	0.6152	0.3061	0.5232	0.0075	0.6331
% max dose	0.1842	0.7350	0.8570	0.2724	0.0605	0.2966

**Table 6 biomedicines-11-01622-t006:** Pooled changes in cardiovascular and haemodynamic parameters by treatment duration for females and males.

Parameter		Females	Males
SBP (mmHg)	MD acute	−35.6 (−64.5; −6.6)	−9.4 (−15.2; −3.7)
	MD sub-acute	−30.8 (−38.5; −23.1)	−21.8 (−28.8; −14.7)
	MD chronic	−19.7 (−20.8; −18.7)	−15.1 (−23.8; −6.4)
DBP (mmHg)	MD acute	−20.4 (−32.2; −8.6)	−6.9 (−9.7; −4.1)
	MD sub-acute	−14.6 (−23.7; −5.5)	−15.1 (−20.0; −10.2)
	MD chronic	−10.9 (−11.5; −10.3)	−10.9 (−16.1; −5.8)

Values are reported as mean difference (MD) compared to baseline with 95% CI. Acute = 0–14 days, sub-acute = 15–30 days, chronic = >31 days, SBP = systolic blood pressure, DBP = diastolic blood pressure.

## Data Availability

No individual patient data are included in this study. Search strategy and results of included papers are presented within the manuscript and are available at the corresponding author upon request.

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
