# Peer review of "Sex Differences in the Anti-Hypertensive Effect of Calcium-Channel Blockers: A Systematic Review and Meta-Analysis"

_biomedicines, 2023, doi:10.3390/biomedicines11061622_

Round 1
Reviewer 1 Report
The meta-analysis is difficult to follow. The authors evaluated the effect of calcium antagonists on blood pressure in the two sexes. The conclusions are that there is a statistically significant reduction only in systolic blood pressure, but not in diastolic
Comments
-Females' blood pressure was much higher under baseline conditions (162 vs 146 mmHG), so the reduction could also be influenced by the initial value
-abstract line 18 calcium antagonists were discovered in 1960 and used for hypertension therapy since 1980. Correct the dates (1945).
-Line 100 studies were excluded if participants received more than one medication as intervention. The authors evaluated many studies but most of the patients studied were from the Melhium study (7400 cases) and in this study 40% of patients were taking a thiazide diuretic
-Line 118 only BP data measured using noninvasive methods. Table 1 includes patients with arterial BP measurement during catheterization.
-Some studies do not refer to hypertensive patients.
-Table 1 reference 38 Gottdien: explain other antihypertensives 185 cases.
-I would remove the two studies that evaluated LVM (line 187) as they refer only to males
-Check references 61-67 that are not given in the text and refer to Kompas
-Page 16 Figure 2 and 6 table shows almost all cases referred to Melhum study 1254 vs 74 cases from the other studies for males, 1672 and 290 females from the other studies. In Table 1, the number of patients is different
In conclusion, the study should be revised considering these discrepancies
Reviewer 2 Report
This work is about a meta-analysis of a large number of studies and a population of 8202 patients using CCB and having sex-stratified data available. The author concluded CCB lowers BP in both sexes, but the decrease in especially SBP is significantly greater in females compared to males. The lowering effect on HR and LVEF is only significant in females. When normalizing blood pressure, differences in these sex-depending effects may be taken into account.
The work provide insight into CCB for HTN and also provided futher research topics about sexual difference.
The author did comprehensive review.
Reviewer 3 Report
The present paper aimed to study the effectiveness of calcium channel blockers [CCB] in acute, subacute and chronic therapy on blood pressure [BP], heart rate [HR] and cardiac function between sexes by performing a systematic review and meta-analysis on studies on CCB from 1945 to May 2020. The study concluded that CCB lower BP in both sexes, but the observed effect is larger in females as compared to males.
A few changes are needed, as follows:
Please explain every abbreviation before using it!
INTRODUCTION: Please also mention sex related effects of smoking on vascular function (Mozos I, et al. Gender Differences of Arterial Stiffness and Arterial Age in Smokers. Int J Environ Res Public Health. 2017 May 26;14(6):565. doi: 10.3390/ijerph14060565). Include a few words about gender-related differences in vascular function in hypertension (Bruno RM, et al. Vascular function in hypertension: does gender dimension matter? J Hum Hypertens. 2023 Apr 15. doi: 10.1038/s41371-023-00826-w).
MATERIALS AND METHODS, Literature search: The aim of the study was related to calcium channel blockers, but you mention (line 71-72 and 77-78) the five antihypertensive drugs …CCB, ACEI, ARB, BB and DIU. Please clarify!
Figure 9: Probably Meta-regression curve of diastolic blood pressure [DBP] not systolic….Please correct!
DISCUSSION should start with your findings!
Study limitations: Please also mention heterogeneity due to different CCB and different durations of therapy.
Round 2
Reviewer 1 Report
no further comment